# MITIGATING CONFLICTS IN MULTI-TASK REINFORCEMENT LEARNING VIA PROGRESSIVELY-TRAINED DYNAMIC POLICY NETWORK

## ABSTRACT

Reinforcement learning is widely applied in various fields, including game playing, robotic control and autonomous driving. However, we find that, when trained for multi-tasking where there exist inter-task conflicts, the standard reinforcement learning algorithm may yield limited performance on individual tasks. To mitigate this, we introduce a dynamic policy network that incorporates diverse computational pathways of varying depths, along with gating modules that selectively activate the appropriate pathways for different tasks. This design, equipped with better flexibility, allows the network to achieve improved multi-task performance. Second, we propose a progressive training technique to mitigate the conflicts among tasks by leveraging proper training order and continual learning techniques. Using the dynamic policy network design and the progressive training technique, we successfully trained a policy capable of performing seven quadrupedal locomotion tasks and a policy that achieved an improved final average reward on ten MiniHack games.

## 1 INTRODUCTION

Reinforcement learning is currently widely applied in various fields, including game playing (K. et al., 2023; W. et al., 2018; S. et al., 2023), robotic control (R. et al., 2021; K. & T., 2021; Luo et al., 2024) and autonomous driving (K. et al., 2021; Elallid et al., 2022; Cao et al., 2022). While reinforcement learning has achieved impressive results across diverse domains, many existing methods (Wang et al., 2024; J. et al., 2024; S. & G., 2021) are tailored to single-task settings, where achieving high multi-task RL performance remains a challenge (Vithayathil Varghese & Mahmoud, 2020). Specifically, if multiple RL tasks are naively trained together, due to conflicts between different tasks, an increase in the performance of one task will commonly hurt the performance of another.

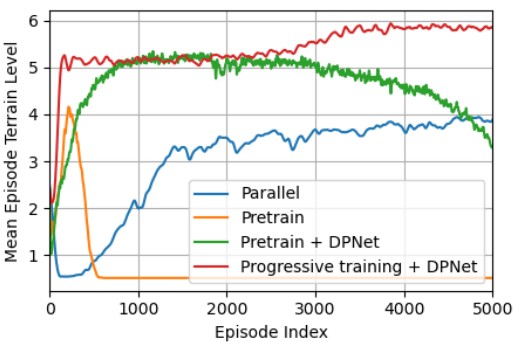

Figure 1: Average training terrain-level curves for quadrupedal robot locomotion tasks under various configurations. Results are averaged across three walking and four parkour tasks.

To illustrate this problem, we took three walking tasks and four parkour tasks that could be trained to gain good performance independently (shown in Section 4.1) and performed naive parallel training on the seven tasks (detailed in Appendix B). The average terrain difficulty level curves during training are shown in Fig. 1, where a higher terrain difficulty level indicates better performance. Under naive parallel training for all walking and parkour tasks (blue curve), the policy reached a limited average terrain difficulty level of approximately 4.0. When the agent underwent pretraining on all walking tasks before being trained concurrently on both walking and parkour tasks (orange curve), a slight performance improvement was observed. However, this approach introduced training instability. The policy initially showed enhanced parkour performance, but at around 150 training episodes, conflicts between the two task types caused a degradation in walking performance, resulting in a significant drop in overall performance.

The primary cause for this negative transfer problem is that the policy network architecture used is ineffective for multi-tasking. A common solution is to propose network architectures that allow more flexible decision making, where different parallel network branches can be selected for different tasks (Shazeer et al., 2017; Yang et al., 2020a), allowing the conflict to be reduced. Building on this idea, we found it also important to increase network decision flexibility by including various pathways with different depths, allowing tasks with different difficulties to be processed by pathways with corresponding depths. To this end, we propose a dynamic policy network (DPNet) design. That is, in the feedforward stage within the policy network, for each feature, we assign multiple auxiliary branches that connect the current feature to both itself and different preceding layer features to incorporate additional computation pathways into the network, allowing decision pathways with different depths to be included. To allow the network to dynamically select pathways with different depths for different tasks, we include gating modules within the auxiliary branches. As shown by the green curve of Fig. 1, by using the DPNet design for the policy network, we obtained a substantial reduction in training instability and an improvement in performance.

Furthermore, contrary to the common belief that parallel training sets the performance upper bound for continual learning (J. et al., 2017; R. et al., 2019), our integration of continual learning with atomic-to-compositional task ordering and action frequency constraints achieved improved performance by mitigating inter-task conflicts. Specifically, with this technique, conflict is avoided by restricting the policy network to generate low-frequency actions suitable for atomic tasks, while performing compositional tasks by imposing high-frequency actions. This deliberate separation of action frequencies avoids conflicts between the two types of tasks and leads to a notable boost in performance. We name this technique *progressive training*. The effect of integrating progressive training and DPNet is shown by the red curve in Fig. 1.

In this work, by training DPNet using progressive training, we successfully trained a single policy network that can simultaneously perform three walking and four parkour tasks. Additionally, we have applied this method to a continual learning benchmark known as MiniHack (Samvelyan et al., 2021), where we obtained improved multi-task performance compared to several baseline RL methods.

## 2 RELATED WORK

**Parallel training with curriculum.** Multi-task reinforcement learning methods typically employ parallel training across tasks, such as parkour (X. et al., 2024), robotic control (J.S. & D., 2024), and autonomous driving (M. et al., 2020). During training, curriculum learning (B. et al., 2009; Chamorro et al., 2024; X. et al., 2024) is often employed to assist the agent to acquire skills, where it typically trains the agent starting from simple to more difficult tasks.

**Continual Reinforcement Learning.** Continual learning is often employed to achieve multi-task reinforcement learning in scenarios where, for example, training data is only available in the future (L. et al., 2024), computing resources are limited (E. et al., 2018), task boundaries are unclear (O et al., 2018), or parallel training is otherwise infeasible. Continual learning research mainly centers on regularization, replay, architectural methods, and hierarchical reinforcement learning. For example, Elastic Weight Consolidation (EWC) (J. et al., 2017) and P&C (S. et al., 2018) are employed to mitigate catastrophic forgetting by preserving the critical weights. SANE (S. et al., 2022) expands the network to learn new tasks and integrates data replay that is similar to CLEAR (R. et al., 2019) to alleviate catastrophic forgetting.

**Combining curriculum learning with continual learning.** In the field of natural language processing, there are already studies that combined curriculum learning with continual learning to improve continual learning performance (Tee & Zhang, 2023) or perform non-compositional expression generation (Zhou et al., 2023). Here we emphasize that, in our proposed progressive training method, instead of treating curriculum learning and continual learning as core methods for improving performance, we use them (together with an additional action frequency constraint) as tools to achieve our objective of restricting the policy to use different action frequency bands to perform different tasks.

**Compositional Networks for Multi-Task Learning.** In the field of multi-task learning, there exist various works that aim for resolving the negative transfer problem (Teh et al., 2017; Wu et al., 2020) by leveraging parallel pathways (Shazeer et al., 2017; Huang et al., 2025; Yang et al., 2020b) or by performing parameter composition (Sun et al., 2022). Moreover, similar to our design, D2R network (He et al., 2024) also leveraged short-cut connections to form decision pathways with different depths

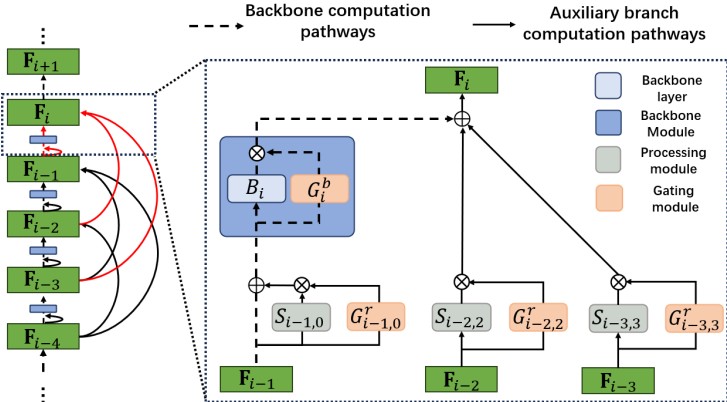

Figure 2: An illustration of applying auxiliary branches (with span $N = 2$) to features $\mathbf{F}_{i-4}, \mathbf{F}_{i-3}, ..., \mathbf{F}_i$ of a the feed-forward backbone of a policy network to form a DPNet. The computation pathways involved in computing $\mathbf{F}_i$ are highlighted in red and the modules involved are visualized in the dotted box.

to gain better multi-task RL performance. However, we wish to point out that, since the D2R design does not expand the policy network as in our DPNet, its learning ability is limited by its backbone.

## 3 METHODS

### 3.1 PRELIMINARIES

**Markov Decision Process.** Reinforcement learning is commonly formulated as the Markov Decision Process (MDP, Sutton & Barto (2018)). An MDP is typically defined as a tuple $\langle \mathcal{S}, \mathcal{A}, \mathcal{R}, \gamma \rangle$, where the agent selects action $a$ based on state $s$, receives a corresponding reward $\mathcal{R}(s, a)$, and transitions to the next state based on the environment dynamics. $\gamma \in [0, 1]$ denotes the discount factor. The overall goal of an MDP is to find an optimal policy $\pi(s, \theta)$ with parameter $\theta$, which is a mapping from states to actions that maximizes the expected cumulative reward over time:

$$\mathcal{R}(\theta, s, \alpha_1, ..., \alpha_K) = \mathbb{E}_{a \sim \pi(\theta)} \left[ \sum_{t=0}^{\infty} \gamma^t \sum_{k=0}^{K} \alpha_k r_{k,t} \right] \tag{1}$$

where $K$ is the number of different rewards $r_{k,t}$ to be maximized to perform the current tasks, $\alpha_k$ is the weight for the $k^{th}$ reward. In this work, the policy $\pi(s, \theta)$ is a neural network with parameter $\theta$.

**Action Frequency.** In this work, we analyze action sequences through a spectral lens. To avoid ambiguity, across the entire paper, the term "action frequency" refers not to an action's rate of occurrence, but to its spectral properties derived via the Discrete Fourier Transform (DFT). That is, in the frequency spectrum obtained with DFT, low-frequency (smooth) actions exhibit dominant energy in low-frequency bands, while high-frequency (abrupt) actions manifest dominant energy in high-frequency bands.

### 3.2 THE DPNET DESIGN

A common approach to improve multi-tasking performance is to design network architectures that incorporate parallel pathways, allowing different pathways to be selectively activated for different tasks (e.g. MoE, Shazeer et al. (2017)), thereby reducing conflicts. However, since different tasks have different difficulties, pathways with a pre-determined depth might be too shallow for some difficult tasks, causing poor decisions to be made. While it might also be unnecessarily deep for some simple tasks, making the capacity of the network to be wasted. Consequently, to enhance multi-task RL capabilities, as illustrated on the left of Fig. 2, we propose to integrate auxiliary branches into the feed-forward backbone layers of the policy network, forming a dynamic policy network (DPNet) with various intertwined parallel decision pathways of different depths. That is, for every backbone layer, we assign $N + 1$ auxiliary branches to it. Within them, $N$ auxiliary branches are shortcuts that connect feature $\mathbf{F}_i$ with preceding backbone layer features $\mathbf{F}_{i-2}, ..., \mathbf{F}_{i-N-1}$, allowing decision pathways with shallower depths (suitable for simpler tasks) to be incorporated. Moreover, we also include a self-dilation auxiliary pathway which maps $\mathbf{F}_{i-1}$ to itself, allowing decision pathways with larger depths (suitable for harder tasks) to be incorporated. In this way, when tasks with different

Figure 3: Conflicts in low-frequency actions between walking and parkour tasks. Gaze visualization of the policy (a) trained only on parkour tasks, (b) trained only on walking tasks, (c) trained on both walking and parkour tasks.

difficulties are encountered, pathways with corresponding depth can be used, allowing the network capacity to be more effectively utilized.

To demonstrate the designs within the DPNet, in the dotted box of Fig. 2, we illustrate how DPNet computes a backbone feature $\mathbf{F}_i$ with its corresponding backbone computation pathway and shortcut auxiliary branches. In each shortcut auxiliary branches, we incorporate two feed-forward modules $S_{i,m}$ and $G_{i,m}^r$. $S_{i,m}$ is processing module included to allow the auxiliary branch to perform feature processing. $G_{i,m}^r$ is gating module with a Sigmoid activation function (M.A. et al., 2019) on its output layer. With this design, its output has values between zero and one, allowing it to dynamically activate the pathway for different tasks. Formally, in each auxiliary branch, both $S_{i,m}$ and $G_{i,m}^r$ process the same input $\mathbf{F}_i$ and compute the auxiliary branch output $\mathbf{M}_{i,m}^r$ by

$$\mathbf{M}_{i,m}^r = S_{i,m}(\mathbf{F}_i) \odot G_{i,m}^r(\mathbf{F}_i), \tag{2}$$

where $\odot$ denotes the element-wise product between two vectors. In the backbone computation pathway, the backbone input $\mathbf{F}_{i-1}$ is first enhanced through a self-dilation auxiliary branch. Formally, the input $\mathbf{F}_{i-1}$ is preprocessed by the computation of $\tilde{\mathbf{F}}_{i-1} = S_{i-1,0}(\mathbf{F}_{i-1}) \odot G_{i-1,0}^r(\mathbf{F}_{i-1}) + \mathbf{F}_{i-1}$. Here, $G_{i-1,0}^r$ controls if the depth increasing self-dilation pathway through $S_{i-1,0}$ is taken. The preprocessed feature $\tilde{\mathbf{F}}_{i-1}$ then flows through the backbone module formed by the backbone layer $B_i$ and a corresponding gating $G_i^b$:

$$\mathbf{M}_i^b = G_i^b(\tilde{\mathbf{F}}_{i-1}) \odot B_i(\tilde{\mathbf{F}}_{i-1}). \tag{3}$$

With the outputs from the shortcut auxiliary branches and the backbone computation pathway, we compute $\mathbf{F}_i$ by

$$\mathbf{F}_i = \mathbf{M}_i^b + \sum_{n=1}^{N} \mathbf{M}_{i-n,n}^r. \tag{4}$$

Additionally, we also propose to initialize the different processing modules ($S_{i,m}$) so that their outputs have near-zero values. With this initialization, if a processing module disrupts the decision of the network, it can be trained to continue to generate near-zero outputs, not interfering with the decision of the rest of the network. Similarly, when initializing the gating modules, we propose to initialize different $\mathbf{G}_i^b$ so that their outputs are close to ones, while initializing different $\mathbf{G}_i^r$ to have output values near zero. With this initialization, when it is disadvantageous to include the computation pathways added by the auxiliary branches, the gating modules can be trained to maintain their initial outputs.

### 3.3 PROGRESSIVE TRAINING

Empirically, we found that when employing our proposed network to optimize both atomic and compositional tasks jointly, a conflict in performance emerges between the two task types. We conjecture that the conflict arises due to two characteristics of these task types:

- For atomic tasks, the inherent simplicity allows the policy network to complete the tasks primarily relying on low-frequency actions (smooth actions). For example, when a quadrupedal robot performs walking, only smooth actions are needed to control its various joints effectively.

- For compositional tasks, to achieve fine-grained control, it requires the policy network to leverage both high-frequency actions (abrupt actions) and low-frequency actions. For

example, when a quadrupedal robot engages in parkour, low-frequency actions facilitate basic walking movements, while abrupt, high-frequency joint adjustments are required for executing delicate actions such as climbing and jumping.

Consequently, conflicts arise when the low-frequency actions required for compositional tasks are different from those needed for atomic tasks.

To provide an intuitive visual example of the conflict, we trained three policies for quadrupedal robot locomotion. The first policy was trained for only parkour tasks (compositional tasks), the second policy was trained for only walking tasks (atomic tasks) while the third was trained with both walking and parkour tasks. During training, we did not impose gaze restrictions and we restricted the policy networks to leverage only low-frequency actions to perform the tasks. The forward walking gaze for the policy networks is shown in Fig. 3. By comparing Fig. 3a and Fig. 3b, it can be seen that, to achieve better parkour performance, the policy in Fig. 3a had a tendency to tilt the head up more intensely, so that it is more convenient to climb and reaches further when crossing gaps. In Fig. 3b, the policy tends to maintain the body of the robot level so that it is easier to move in all directions. This demonstrates that a conflict exists in low-frequency action (the gaze of the quadrupedal robot). As a consequence, when we naively trained the walking and parkour tasks in parallel, due to the conflict, the policy obtained a poor gaze, causing the robot to crawl forward (Fig. 3c).

To mitigate this issue, we propose a progressive training technique. This training technique encourages the policy network to generate low-frequency actions appropriate for atomic tasks while performing compositional tasks by applying high-frequency actions on top of them. We achieve this goal by properly combining continual learning restriction, atomic-to-compositional training and action frequency constraints (smoothness rewards).

Specifically, in the early stages, the policy network is trained in atomic tasks restricted with a strong smoothness reward (reward that penalizes policy output acceleration (Haarnoja et al., 2018; Raffin et al., 2022)) to acquire the corresponding low-frequency actions. That is, we optimize the policy network parameter $\theta$ by

$$\text{argmax}_\theta \, \mathbb{E}_{s \sim P, a \sim \pi(\theta)} \left[ \mathcal{R}(\theta, s, \alpha_a, \alpha_{\text{smooth}}) \right], \tag{5}$$

where $\alpha_a$ is the weight for the atomic task rewards, $\alpha_{\text{smooth}}$ is the weight for action smoothness reward. When performing subsequent training on compositional tasks, we apply continual learning techniques (such as EWC (J. et al., 2017) or CLEAR (R. et al., 2019)) to the policy network so that we constrain its low frequency actions learned from atomic tasks to be unchanged while training the policy network to perform the compositional tasks. During compositional task training, we reduce the strength of the smoothness reward. That is, we optimize the policy network parameter $\theta$ by maximizing

$$\text{argmax}_\theta \, \mathbb{E}_{s \sim P, a \sim \pi(\theta)} \left[ \mathcal{R}(\theta, s, \alpha_a, \alpha_c, \epsilon \alpha_{\text{smooth}}) \right] - L_c(\theta), \tag{6}$$

where $\epsilon$ is an empirically selected value smaller than 1, $\alpha_c$ is the weight for compositional task rewards, $L_c$ is the loss function for continual learning. With this design, given that the low-frequency actions remain relatively unchanged, while the high-frequency actions are modifiable, the policy is forced to perform the compositional tasks by applying high-frequency actions onto the low-frequency actions learned for the atomic tasks.

**Distinguishing Compositional Tasks from Atomic Tasks.** Since compositional tasks often require pretraining on atomic tasks for efficient learning (O et al., 2018; M. et al., 2022), we empirically identified direct training reward as an effective metric for differentiation. That is, within the same type of task, those tasks that yield higher rewards through direct training tend to be atomic, whereas tasks with lower rewards are more likely to be compositional.

## 4 EXPERIMENTS

**Quadrupedal Robot Locomotion Task Environment.** In our experiments, we leveraged the Isaacgym environment (Makoviychuk et al., 2021) to evaluate our design on quadrupedal robot locomotion tasks. We included seven tasks, three were categorized as walking tasks, while the other four were categorized as parkour tasks. For the walking tasks, we trained the policy to control the robot to walk on three terrains, namely flat ground, hills and stairs (Rudin et al., 2021). For the parkour tasks, we trained the robot to walk forward to cross four types of terrains, the gap terrain, the box climbing terrain, the hurdle terrain and the tilted ramp terrain (X. et al., 2024). Visualization on terrains in our experiments is included in Appendix B.

**Quadrupedal Robot Locomotion Task Training Configuration.** Within our experiments, we used the existing walking (Rudin et al., 2021) and parkour (X. et al., 2024) curricula and integrated our progressive training technique. For progressive training, we treated the walking tasks as atomic tasks, while treating the parkour tasks as compositional tasks since the robot must walk properly to perform parkour. So we divided the training into two major stages. In the first stage, we trained the robot to walk in all directions with randomly selected linear and angular velocity. The training was performed on the three walking terrains in parallel. In this stage, a smoothness reward (Tan et al.) was applied. In the second stage, we removed the smoothness reward, added EWC and trained the agent on walking and parkour terrains in parallel. In this stage, we maintained a 4-to-6 actor number ratio between walking and parkour training. In all stages, we included 4500 agents in the Isaacgym environment to perform training in parallel. More training configuration details are described in Appendix B.

**Quadrupedal Robot Locomotion Task Policy Network Architecture.** For the policy network in quadrupedal robot locomotion tasks, we used a multi-layer perceptron architecture. Specifically, we leveraged a simple MLP backbone in the first training stage and added the auxiliary branches at the beginning of the second stage to make the policy network a DPNet. The detailed architecture configuration is described in Appendix B. We leveraged a lightweight version (described in Appendix C) of the gating module and set the span parameter $N$ of the auxiliary branch to 3.

**Evaluation Metric for Quadrupedal Robot Locomotion Tasks.** To evaluate the performance of a quadrupedal robot control policy on walking and parkour, following the convention in Extreme Parkour (X. et al., 2024), we evaluated the average terrain level reached (Avg.TLR) of the policy across the seven types of terrains. To calculate the TLR on a certain kind of terrain, we initialize a matrix of $3 \times 10$ terrains. In each column, the terrains have the same difficulty but with different randomly initialized details. For different columns, the terrain difficulty increases from level 1.0 to 10.0 as the column index increases. Then we initialize around 250 agents on the terrains and find the highest difficulty level where 90% of agents could cross more than 75% of the terrain's length. We define this difficulty level as the TLR of the current policy on the current kind of terrain. The Avg.TLR is the average of TLR across all seven terrains. We averaged all results over three trials.

**MiniHack Environment.** MiniHack is built upon the NetHack learning environment (K. et al., 2020), which serves as one of the training environments for our experiments. Specifically, we utilize the navigation tasks within MiniHack. The MiniHack navigation task challenges the agent to reach a designated goal location while navigating various obstacles, including avoiding monsters and traversing intricate mazes. We selected 10 distinct tasks from the MiniHack navigation environment for training. The agent performs actions such as searching, door opening and eating based on environmental conditions. A reward is granted when the agent successfully reaches the goal. Otherwise, the agent receives either no reward or a penalty. The destination can only be reached when certain conditions are fulfilled. Please refer to Appendix D for a detailed description of the MiniHack tasks involved.

**Experiment Configuration in MiniHack.** In terms of policy network architecture, we leveraged a convolutional neural network (CNN) architecture as our backbone and included auxiliary branches with a span of 2 to make the policy network a DPNet. In addition, we adopted the IMPALA method (E. et al., 2018) to perform training and integrated our progressive training technique. During training, we iterated through 10 tasks (Room-Random-5x5-v0, Corridor-R2-v0, Room-Dark-5x5-v0, Corridor-R3-v0, Room-Monster-5x5-v0, CorridorBattle-v0, Room-Trap-5x5-v0, HideNSeek-v0, Room-Ultimate-5x5-v0, HideNSeek-Lava-v0, described in Appendix D) twice during training. We have also set $\alpha_a$ to be 0, since we found that without explicit smoothness restriction, under the current training order, the policy naturally generated low-frequency actions for atomic tasks. Detailed architecture and training parameter configuration are described in Appendix D.

**Evaluation Metric in MiniHack.** The final average reward $R_{\text{final}}$ across all tasks is used as the evaluation metric for policy performance. The specific calculation process is as follows:

$$R_{\text{final}} = \frac{1}{K}(R_{1,\text{final}} + R_{2,\text{final}} + \cdots + R_{K,\text{final}}), \tag{7}$$

where $K$ is the number of tasks, $R_{K,final}$ is the final reward of the $K^{th}$ task. In MiniHack, we performed all experiments three times to calculate the average and standard deviation of the results.

Table 1: Evaluating the performance of our method against task-specific policies and policies trained with parallel training. TLR of 10.0 indicates that a policy reached the highest difficulty level on the task. Config. indicates RL algorithm configuration.

| Config. | Flat | Stair | Slop | Gap | Climb | Hurdle | Tilted Ramps | Avg. TLR |
|---|---|---|---|---|---|---|---|---|
| Parkour | – | – | – | 9.7±0.5 | 10.0±0.0 | 10.0±0.0 | 8.7±0.8 | – |
| Walking | 10.0±0.0 | 9.3±0.5 | 10.0±0.0 | – | – | – | – | – |
| Parallel | 1.0±0.0 | 1.0±0.0 | 1.0±0.0 | 5.7±1.2 | 5.7±0.8 | 6.7±0.8 | 5.7±0.5 | 3.8±0.3 |
| Pretrain | 9.7±0.5 | 5.0±0.0 | 9.7±0.5 | 4.3±0.5 | 9.0±0.0 | 8.0±0.0 | 7.3±0.5 | 7.6±0.1 |
| Ours | 10.0±0.0 | 9.0±0.8 | 10.0±0.0 | 8.0±0.8 | 10.0±0.0 | 9.7±0.5 | 8.0±0.0 | **9.2±0.1** |

Table 2: Ablation study on different components. B-Gating and A-Gating indicates the presence of gating module on backbone layers and auxiliary branches respectively. A2C indicates atomic-to-compositional training, Smooth indicates presence of the smoothness rewards during training.

| Train. Type | Arch | Shortcut | Self-dilation | B-Gating | A-Gating | A2C | EWC | Smooth | Avg.TLR |
|---|---|---|---|---|---|---|---|---|---|
| Pretrain | MLP | - | - | - | - | ✓ | - | - | 7.6±0.1 |
| Pretrain | DPNet | ✓ | ✓ | ✗ | ✗ | ✓ | - | - | 7.8±0.1 |
| Pretrain | DPNet | ✓ | ✓ | ✓ | ✗ | ✓ | - | - | 7.9±0.1 |
| Pretrain | DPNet | ✓ | ✗ | ✓ | ✓ | ✓ | - | - | 7.8±0.1 |
| Pretrain | DPNet | ✓ | ✓ | ✓ | ✓ | ✓ | - | - | 8.0±0.1 |
| Progressive | DPNet | ✓ | ✓ | ✓ | ✓ | ✓ | ✓ | ✗ | 8.6±0.1 |
| Progressive | DPNet | ✓ | ✓ | ✓ | ✓ | ✓ | ✓ | ✓ | 9.2±0.1 |

## 4.1 COMPARISON AGAINST PARALLEL TRAINING

To demonstrate that our method is effective for boosting multi-task RL performance under a parallel training setting, we evaluated our method on the three walking and four parkour tasks in the Isaacgym environment.

As demonstrated in Table 1, by performing naive parallel training across the seven locomotion tasks with an MLP policy network architecture, a low Avg.TLR of 3.8 was obtained. Specifically, it failed to perform walking-related tasks. By introducing pretraining on walking, then performing parallel training on walking and parkour, the walking performance of the policy increased, obtaining an Avg.TLR of 7.6. In contrast, by training a DPNet with the progressive training technique (Ours in Table 1), an Avg.TLR of 9.2 was obtained. Specifically, it can be seen that our model had a performance comparable to models trained specifically for walking and parkour on all tasks. That is, our method successfully achieved multi-tasking.Demo videos are included in our supplementary material.

## 4.2 DESIGN ANALYSIS

**Ablation Study.** Table 2 presents an ablation study examining the impact of various design choices. As a baseline, an MLP policy network trained in parallel across the seven quadrupedal robot tasks (preceded by walking pretraining) yielded an average TLR of 7.6. Next, we integrated the DPNet design into the configuration. Specifically, by sequentially incorporating auxiliary branches (without gating), gating modules for the backbone layers, and gating modules for the auxiliary branches, the average TLR improved incrementally from 7.6 to 7.8, 7.9 and 8.0. Moreover, when the self-dilation auxiliary branches were removed, the average TLR fell from 8.0 to 7.8, demonstrating the importance of enabling the DPNet to choose deeper pathways. Furthermore, by employing EWC and the smoothness reward from progressive training on top of atomic-to-compositional training (pretraining followed by parallel training), the average TLR of the DPNet increased from 8.0 to 8.6 and 9.2 respectively. This demonstrates that each component within progressive training has a contribution to the final performance.

**Effect of Auxiliary Branch Span $N$.** Empirically, we found it important to set a large value for $N$, since it provides more pathways with different depths for the gating modules to select. Specifically, under progressive training, when we decreased the value of $N$ from 3 through 2, 1, to 0 (when $N = 0$, we removed auxiliary branches for both shortcut and self-dilation modules), the Avg.TLR of the policy network decreased from 9.2 to 8.7, 6.4 and 6.3.

Table 3: Comparing performance of DPNet against different multi-task RL architecture under progressive training.

| Model | MoE | MoE-Loco | D2R | Soft-Modul. | PACO | CARE | DPNet |
|---|---|---|---|---|---|---|---|
| Avg.TLR | 8.6±0.2 | 7.3±0.1 | 6.6±0.3 | 8.4±0.2 | 7.8±0.1 | 6.8±0.1 | **9.2±0.1** |

**Visualizing DPNet Effective Decision Depth.** To demonstrate that with the auxiliary branch design, the policy network can select different pathways with different depths on different tasks, we compared the effective decision depth (see Appendix F for its computation) between parkour and walking tasks. We collected the effective decision depth from 250 agents on 25 random terrains per task type, with difficulties ranging from 1.0 to 10.0. For each agent, we recorded the effective decision depth of 200 action steps. The boxplots of the collected results are shown in Fig.4. The median effective decision depth for parkour tasks (5.88) was higher than walking tasks (5.03). Additionally, the first quartile (5.33) and the lower whisker (4.88) of the parkour tasks were substantially higher than the corresponding values for walking tasks (4.49 and 3.05, respectively). These

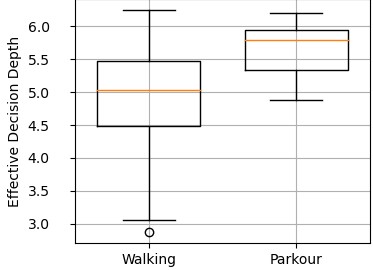

Figure 4: Boxplot for effective decision depth of DPNet on walking and parkour tasks.

results confirm that the DPNet utilizes deeper pathways for more complex compositional tasks and shallower pathways for the simpler atomic tasks, demonstrating its ability to adaptively select decision pathways based on task demands. The boxplot for the effective decision depth of DPNet on the seven individual tasks is analyzed in Appendix F.

**DPNet Action Frequency Spectrum for Atomic and Compositional Tasks.** Under progressive training, we analyzed if DPNet leveraged actions from different frequency bands to perform atomic (walking) and compositional (parkour) tasks in our locomotion environment. In particular, we deployed 250 DPNet controlled agents on 25 random parkour terrains (difficulty levels 1.0–10.0), ran the agents for 800 simulation steps and computed their action frequency spectrum via Fourier transform (details in Appendix E). Similarly, we also computed the action frequency spectrum for walking terrains. Figure 5 visualizes the resulting frequency spectrum. In the walking (atomic) scenario, the spec-

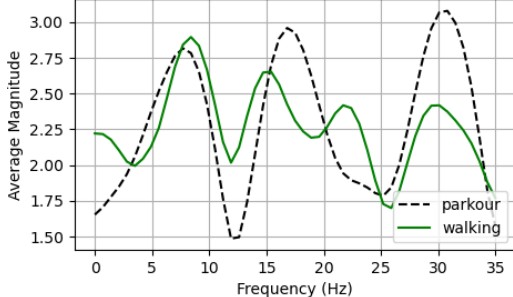

Figure 5: Policy action frequency spectrum of atomic (walking) tasks and compositional (parkour) tasks.

trum exhibits a single dominant peak around 8 Hz, whereas in the parkour (compositional) tasks we observe pronounced peaks at both low and high frequencies. This demonstrates that with progressive training, the DPNet is trained to use actions from different frequency bands to perform different tasks.

### 4.3 COMPARISON AGAINST EXISTING MULTI-TASK RL ARCHITECTURES

To evaluate DPNet against existing baseline architectures, we trained six existing multi-task RL architectures under progressive training and under comparable parameter count and FLOPs (both around 9.5M). The only exception was that, D2R failed to train under large parameter count, so we leveraged the largest version (with a parameter count and FLOPs of 0.5M) that successfully learned to perform the quadrupedal locomotion tasks. The results are presented in Table 3. Specifically, DPNet had an Avg.TLR of 9.2. In comparison, MoE (Shazeer et al., 2017), Soft-Modulation (Yang et al., 2020a) and PACO (Sun et al., 2022) obtained relative high Avg.TLRs of 8.6, 8.4 and 7.8 respectively. But for MoE-loco (Huang et al., 2025), D2R (He et al., 2024) and CARE (Sodhani et al., 2021), even under progressive training, without an architecture design that is sufficiently effective, they obtained relative low Avg.TLRs of 7.3, 6.6 and 6.8 respectively.

### 4.4 EVALUATION IN MINIHACK

**Analysis on the Influence of Training Order.** We conducted three experiments with different training orders and plotted the reward curves of different tasks (Fig.6). The corresponding average reward curves are also plotted to better visualize how training order influences conflict avoidance. The correspondence between the ID of different tasks, the name of the task and their direct training reward is illustrated in Table S2. In experiment (a), by ordering different tasks by their type (training

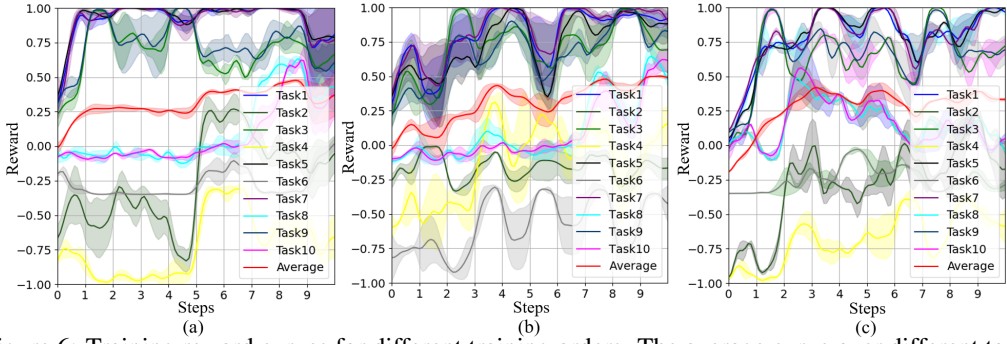

Figure 6: Training reward curves for different training orders. The average curve over different tasks is highlighted in red. All tasks were iterated once. Reward curves for ordering tasks by (a) type, (b) atomic-to-compositional, (c) compositional-to-atomic.

Table 4: Comparison of final average rewards of different continual learning methods on MiniHack.

| Methods | EWC (J. et al., 2017) | P&C (S. et al., 2018) | CLEAR (R. et al., 2019) | SANE (S. et al., 2022) | Ours |
|---|---|---|---|---|---|
| Final Avg. Reward | 0.445±0.001 | 0.405±0.003 | 0.538±0.005 | 0.502±0.003 | **0.578**±0.001 |

Room-related tasks first, then Corridor- and HideNSeek- related tasks), we obtained an task order of 1, 3, 5, 7, 9, 2, 4, 8, 10, 6. During training the average reward initially increased but dropped after training the ninth task (indicating a conflict), resulting in a final reward value of approximately 0.40. Then, in experiment (b) we leveraged an atomic-to-compositional order (of 1, 6, 7, 4, 3, 2, 5, 8, 9, 10) obtained by first ranking the tasks within each type with decreasing direct training reward, then form the overall sequence by interleaving the tasks from different types. As training progressed, the average reward curve generally demonstrated an increasing trend, resulting in a higher final reward value of 0.46. In experiment (c), we leveraged the reversed compositional-to-atomic training order. As training progressed, the average reward curve first increased, then maintained at a steady level of around 0.30, resulting in a low final reward of around 0.36. This confirms that an atomic-to-compositional training order is important for resolving conflicts.

**Comparison Against Continual Learning Baselines.** Considering that continual learning is an important technique for achieving multi-task RL, we compared our method with existing baselines in MiniHack. We performed all experiments three times and reported the average and standard deviation of the final average rewards. All policies were trained under identical task order and the task sequence was iterated twice to train the models thoroughly. Table 4 shows a comparison of the final average rewards of different continual learning methods. Specifically, the EWC method obtained a final average reward of 0.445, which was 0.133 lower than our method that obtained an average final reward of 0.578. Similarly, the average final reward of our method was also higher than the final average rewards of P&C, CLEAR and SANE. The final rewards of different individual tasks are detailed in Appendix H.

## 5 LIMITATION

Our progressive training method uses direct rewards to identify atomic and compositional tasks, but because these rewards are not comparable across task types, human expertise is needed to properly interleave task sequence from different types together to form the overall sequence. This mirrors complex human learning on tasks such as parkour or badminton, where instructors are required to design appropriate progressions. Given this, we find this limitation to be acceptable.

## 6 CONCLUSION

In this work, to improve multi-task RL performance, we propose two designs. First, the DPNet design allows the policy network to dynamically select decision pathways with different depths on different tasks, enhancing the network's flexibility for multi-task RL. On the other hand, with progressive training, we allowed the DPNet to better avoid conflicts between different tasks during training, leading to improved final performance. Our method is also applicable across tasks and training frameworks.

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

## A  DECLARATION ON LARGE LANGUAGE MODEL USAGE

When writing this paper, we used large language models (ChatGPT and DeepSeek) to polish the grammar and writing. In addition, we employed these models to assist with literature search. Specifically, we leveraged DeepSeek's search capabilities to identify prior research that integrates curriculum learning with continual learning (Tee & Zhang, 2023; Zhou et al., 2023).

## B  DETAILS ON QUADRUPEDAL ROBOT EXPERIMENTS

**The Seven Tasks Included.** Fig. S1 a, b, c, d and e, demonstrate the flat ground, hill (climbing), hill (descending), stairs (climbing) and stairs (descending) terrains. On those terrains, the quadrupedal robots were initialized at the middle of the terrain and were commanded to move in random directions with random angular velocity commands. Moreover, in Fig. S1 f, g, h and i,the gap crossing, box climbing, the hurdle and tilted ramp terrains are demonstrated. On those terrains, the quadrupedal robots were initialized on the platforms located in front of the parkour terrains. The robots were commanded to move forward across the parkour terrains. As demonstrated in Fig. S1 j, in the second sub-stage of the walking training stage, we also modified the walking terrains to include gaps so that the gaze of the robot could be pretrained for performing parkour. Here, we only demonstrate the modified version of the stair terrain, but identical gaps were also added on flat and hill terrains.

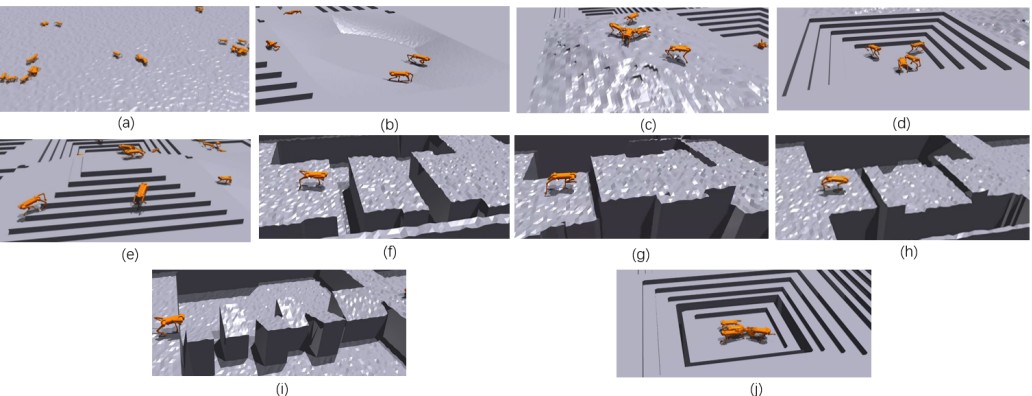

Figure S1: Different terrains included in the quadrupedal robot experiments. (a), (b), (c), (d) and (e) illustrate the flat ground, hill climbing, hill descending, and stairs terrains, stairs climbing and stairs descending terrains. (f), (g), (h) and (i) illustrate the gap crossing, box climb, hurdle and tilted ramp terrains. (h) illustrates the stair terrain added with gaps.

**Training Configuration Details.** In practice, we further divided the two training stages, each stage containing two sub-stages. In the first sub-stage of the first stage, we trained for all three walking tasks and in the second sub-stage, we continued to train for the walking tasks but leveraged the terrains with gaps added (illustrated in Fig. S1h) to pre-train for parkour. In the second stage, we included gap crossing and box climbing tasks in the first sub-stage. In this sub-stage, 50% of terrains were set to gap crossing and 50% of terrains were set to climbing. In the second sub-stage, we further added training for hurdle and tilted ramps, where we set 25% for each type of terrain. Within the second stage, we also included weight decay to increase network plasticity (Dohare et al., 2024).

**Network Architecture for Quadrupedal Robot Policy Network.** In the quadrupedal robot experiments, we leveraged the PPO algorithm to perform reinforcement learning. Within the PPO algorithm, an actor network and a critic network is usually included. Fig. S2 illustrates the design of the actor network we used in our experiments, where it adopted an MLP architecture with two stages. The first stage contains two modules, the first module was a scandot (vision) encoder, which took in five sets of historical and one set of present scandots of the terrain in front of the robot to encode scandots into a feature vector. The historical scandots were taken with an interval of 6 steps relative to the present scandot. This module had two hidden layers with hidden dimensions of 64 and 20. The second module was a past action encoder. It took 8 sets of historical joint positions of the robot (each with 16 values) and encoded them into a past actions feature vector with 32 dimensions.

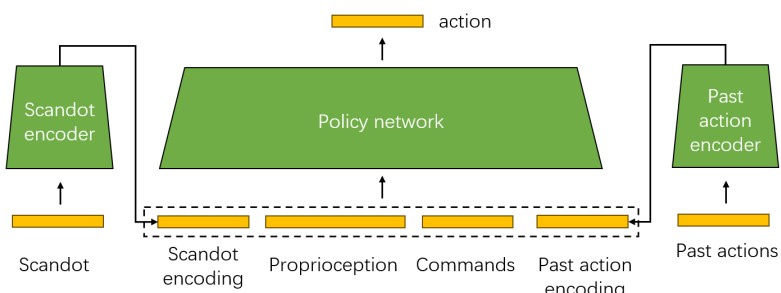

Figure S2: The MLP-based actor network for the policy of the quadrupedal robot.

The encoding was performed with an MLP with a single hidden layer, where the hidden dimension was 64. The second stage was the policy network, where it took in the scandot encoding, direction commands, the past actions encoding and the inertial measurement unit (IMU) information. With the inputs provided, the second block could predict an action. The policy network had 5 hidden layers, with hidden dimensions of 402, 280, 295, 136 and 94. The critic network used in our research had an identical architecture to the actor network, except it had an output layer for generating value predictions. In our experiments, we only added the auxiliary branches to the policy block of the actor network. In the auxiliary branches, we set processing modules to be MLPs with three layers, where the hidden dimension was identical to the dimension of the feature vector to which it is routed. We used the lightweight gating described in Appendix C, where the gating modules were two-layer MLPs with hidden dimensions of 80 and 40 at different depths.

## C   Lightweight Auxiliary Branch

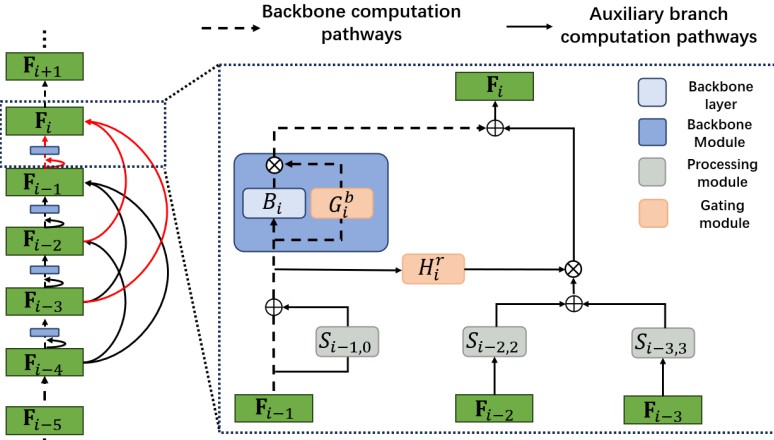

Figure S3: An illustration of applying lightweight auxiliary branches (with span $N = 2$) to features $\mathbf{F}_{i-4}, \mathbf{F}_{i-3}, \mathbf{F}_{i-2}, \mathbf{F}_{i-1}$ and $\mathbf{F}_i$ of a feed-forward backbone to form an DPNet. The computation pathways involved in computing $\mathbf{F}_i$ are highlighted in red and the modules involved are visualized in the dotted box.

Considering that the auxiliary branch design described in the main paper might be computationally prohibitive for some tasks, we also proposed a lightweight version. That is, instead of assigning a gating module for each $S_{i,m}$, we assigned a gating module $H_i^r$ for all shortcut auxiliary branch outputs mapped to each backbone layer. Specifically, as demonstrated in Fig. S3, $H_i^r$ takes the same input as its corresponding backbone layer and modulates all shortcut module outputs mapped to the current backbone layer simultaneously. Formally, we let the input to $H_i^r$ be $\mathbf{F}_{i-1}$ modulated by $S_{i-1,0}$:

$$\tilde{\mathbf{F}}_{i-1} = S_{i-1,0}(\mathbf{F}_{i-1}) + \mathbf{F}_{i-1} \tag{8}$$

and we let $\mathbf{M}_i^r$, the shortcut auxiliary branch output modulated by $H_i^r$ be

$$\mathbf{M}_i^r = \sum_{n=1}^{N} S_{i-n,n}(\mathbf{F}_{i-n}) \odot H_i^r(\tilde{\mathbf{F}}_{i-1}). \tag{9}$$

The computation for obtaining $\mathbf{M}_i^b$ remains unchanged, but to obtain $F_i$, we perform the computation of

$$\mathbf{F}_i = \mathbf{M}_i^b + \mathbf{M}_i^r. \tag{10}$$

With this design, although we have reduced the number of gating modules, but with remaining ones, different pathways with different depths can still be selected, making the policy network a dynamic policy network.

## D   MINIHACK TASKS AND EXPERIMENT DETAILS

**The Ten Tasks Included.** We selected 10 tasks from the MiniHack game environment as training tasks, including 1.Room-Random-5x5-v0, 2.Corridor-R2-v0, 3.Room-Dark-5x5-v0, 4.Corridor-R3-v0, 5.Room-Monster-5x5-v0, 6.CorridorBattle-v0, 7.Room-Trap-5x5-v0, 8.HideNSeek-v0, 9.Room-Ultimate-5x5-v0, 10.HideNSeek-Lava-v0. We present examples of the initial observations that an agent may encounter for each task in the MiniHack benchmark in Fig. S4 and the detailed descriptions on the tasks follow below.

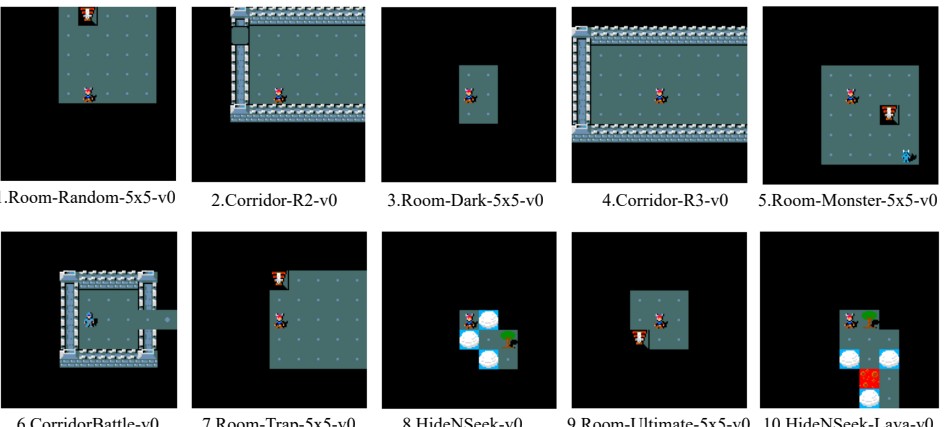

Figure S4: Examples of initial observations for each task in the 10-task MiniHack sequence.

**1.Room-Random-5x5-v0:** Explore the randomly generated room to reach the goal. The layout, player, and goal positions are random in each episode.

**2.Corridor-R2-v0:** Reach the exit by navigating through two connected corridors. The player and exit positions are random in each episode.

**3.Room-Dark-5x5-v0:** Find the goal hidden in the dark room. The player's position and the goal location are random in each episode.

**4.Corridor-R3-v0:** Reach the exit by navigating through three connected corridors. The player and exit positions are random in each episode.

**5.Room-Monster-5x5-v0:** Reach the goal while avoiding or defeating the monster in the room. Player, monster, and goal positions are random in each episode.

**6.CorridorBattle-v0:** Fight monsters in the corridor and through the corridor to reach the exit. Player, enemies, and exit positions are random in each episode.

**7.Room-Trap-5x5-v0:** Reach the goal while avoiding hidden traps scattered in the room. Player and goal positions are random in each episode.

**8.HideNSeek-v0:** Find and reach the hidden target while avoiding detection. Player and goal positions are random in each episode.

**9.Room-Ultimate-5x5-v0:** Reach the goal while navigating through a room filled with both monsters and traps. Player, monsters, traps, and the goal are random in each episode.

**10.HideNSeek-Lava-v0:** Find and reach the hidden target while avoiding dangerous lava hazards. The target's position and the lava are random in each episode.

**Training Configuration Details.** In our experiments, we trained all RL agents in the MiniHack environment for 2 epochs using the IMPALA(E. et al., 2018)-based training framework (unless otherwise stated). In each epoch, EWC(J. et al., 2017) was incorporated to facilitate continual learning across all tasks. Each task was trained for 1e6 steps, and during training, all tasks were evaluated every 1e5 steps to record the reward performance. The training hyperparameters of our proposed method on MiniHack are shown in Table S1.

Table S1: The hyperparameters of our proposed method in MiniHack tasks.

| Hyperparameters | Ours |
| --- | --- |
| Num. actors | 16 |
| Learner threads | 2 |
| Batch size | 25 |
| Unroll length | 20 |
| Grad clip | 40 |
| Reward clip | [-1,1] |
| Entropy cost | 0.01 |
| Discount factor | 0.99 |
| Learning rate | 3e-4 |
| EWC $\lambda$ | 100 |
| EWC, min. task steps | 1e6 |
| Fisher samples | 100 |
| Normalize Fisher | No |
| Replay buffer size | 1e6 |

**Network Architecture for MiniHack Policy Network.** For the actor network architecture, we utilized the CNN framework illustrated in Fig. S5. Specifically, we leveraged a CNN backbone consisting of three CNN blocks and two MLP layers, each CNN block was connected with auxiliary branches. For each backbone CNN block, the corresponding processing module within the auxiliary branch contained two convolutional layers, while the gating module comprised two MLP layers followed by a sigmoid activation function. As demonstrated by Fig. S5, the dimension of the MiniHack game image was $1 \times 3 \times 84 \times 84$. In the $\text{CNNBlock}_1$, we added a maxpooling layer with kernel size 3 and stride 2 between the CNN layers in $\text{CNNBlock}_1$ to reduce the feature size. The dimensions of hidden features $F_1, F_2, F_3$ after CNN Blocks were $1 \times 32 \times 42 \times 42$. The auxiliary branch pathways we leveraged are illustrated in Fig. 2b. Specifically, the span of the auxiliary branches was 2 and we have only included auxiliary branches for the CNN blocks. After the first MLP layer, the $F_4$ computed was a vector of dimension 512. The action prediction was obtained after the last MLP layer. The critic network used in our research had an architecture identical to the actor network, except that it had an output layer for generating value predictions.

# E  ACTION FREQUENCY SPECTRUM COMPUTATION

To compute the action frequency spectrum of a task, we executed the policy in the environment and collected its output action curves. That is, we collected an action sequence matrix $\mathbf{M} \in \mathbb{R}^{N \times K \times T}$ where

- $N$ was the number of parallel sequences (samples),

- $K$ was the number of action outputs for the policy network, and

- $T$ was the length (number of time steps) of each sequence.

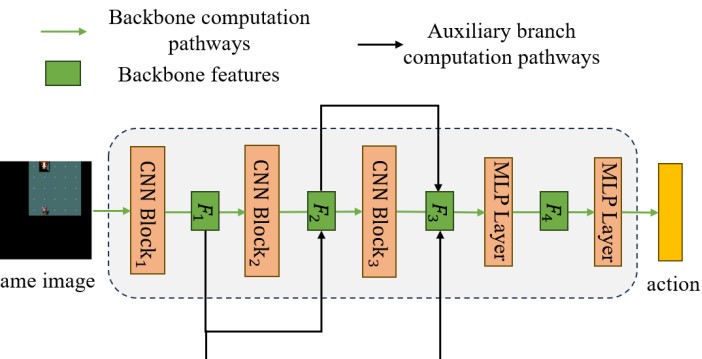

Figure S5: The CNN architecture-based actor network for the policy of the MiniHack tasks.

In our implementation, we performed a real-valued fast Fourier transform (rFFT) along the temporal dimension. To do so, the FFT was computed with a fixed transform length $n$ (we set $n = 100$) so that the number of frequency components was

$$F = \left\lfloor \frac{n}{2} \right\rfloor + 1. \tag{11}$$

For each sequence $n \in \{1, \ldots, N\}$ and for each action $k \in \{1, \ldots, K\}$, we defined the discrete Fourier transform (DFT) of the time signal as

$$\hat{M}_{n,k,f} = \sum_{t=0}^{n-1} M_{n,k,t} \exp\left(-\frac{2\pi i\, f\, t}{n}\right), \quad \text{for } f = 0, 1, \ldots, F - 1. \tag{12}$$

Since the input was real-valued, we used the rFFT, which returned only the non-negative frequency components.

Next, the magnitude of the Fourier coefficients was computed as

$$M'_{n,k,f} = \left| \hat{M}_{n,k,f} \right|, \tag{13}$$

which represented the absolute value (i.e., amplitude) of the frequency component $f$.

To obtain a robust frequency representation for each action, we averaged the magnitudes over all $N$ sequences:

$$P_{k,f} = \frac{1}{N} \sum_{n=1}^{N} M'_{n,k,f}. \tag{14}$$

This yielded the per-action frequency distribution, with

$$\mathbf{P} \in \mathbb{R}^{K \times F}, \tag{15}$$

where the element $P_{k,f}$ corresponds to the average magnitude of the frequency component $f$ for action output $k$ of the policy network.

To compute the overall action frequency spectrum for a task, we need to aggregate across the $K$ action dimensions by summing over the rows of $\mathbf{P}$:

$$d_f = \sum_{k=1}^{K} P_{k,f}, \quad \text{for } f = 0, 1, \ldots, F - 1.$$

This resulted in a task-level frequency distribution,

$$\mathbf{d} \in \mathbb{R}^F,$$

where each element $d_f$ represents the total average amplitude of frequency component $f$ across all action outputs. This final representation captures the overall frequency characteristics of the policy behavior for the given task type and we plotted it as the action frequency spectrum.

## F  COMPUTATION OF EFFECTIVE DECISION DEPTH

This section details the method for computing the *effective decision depth* (EDD) of a forward pass through the Dynamic Policy Network (DPNet). The effective decision depth is a metric that quantifies the cumulative processing depth for an input as it is propagated through the network.

Effective decision depth (EDD) is computed in a single bottom-up sweep through the network. Starting from the input feature $\mathbf{F}_0$ (with EDD initialized to zero), we proceed layer by layer up to the final feature $\mathbf{F}_L$. At each layer $i$, we consider all pathways originating from earlier features $\mathbf{F}_{i-n}$ and compute the depth as the sum of the EDD of the source feature and the EED of the connecting module. If the module is gated, its EED is given by the mean gate activation multiplied by the module's depth. If the module is not gated, its cost is computed as the mean absolute activation of the module. The EDD of $\mathbf{F}_i$ is then obtained by averaging the EDD of all incoming pathways. Repeating this process for $i = 0$ through $L$ yields the final value $C_L$, which represents the overall effective decision depth of the network.

Formally, we let $C_i$ denote the effective decision depth at the output of the $i^{\text{th}}$ backbone layer. The computation proceeds as follows:

1. **Initialization:** The effective decision depth for the initial input feature, $C_0$, is initialized to 0.

2. **Iterative Update:** For each subsequent backbone layer $i$, the effective decision depth $C_i$ is computed based on the previous $N$ depths $\{C_{i-1}, C_{i-2}, ..., C_{i-N}\}$ and the activations of the layer's components connected to the current backbone layer output. The update is performed in three steps:

   - **Base Depth Contribution:** A base depth value, $\tilde{C}_i^b$, is calculated by incorporating contributions from the current backbone layer and its self-dilation auxiliary branch:

     $$\tilde{C}_i^b = C_{i-1} + \mathbb{E}[\mathbf{G}_i^b] \cdot D_i^b + \mathbb{E}[|\mathbf{S}_{i,0}|] \cdot D_{i,0}^s \tag{16}$$

     where:
     - $\mathbb{E}[\mathbf{G}_i^b]$ is the mean output of the backbone gating module at layer $i$
     - $D_i^b = 1$ is the depth of the current backbone layer
     - $\mathbb{E}[|\mathbf{S}_{i,0}|]$ is the mean absolute output of the self-dilation processing module
     - $D_{i,0}^s$ is the depth of the self-dilation processing module

     The products $\mathbb{E}[\mathbf{G}_i^b] \cdot D_i^b$ and $\mathbb{E}[|\mathbf{S}_{i,0}|] \cdot D_{i,0}^s$ estimate the effective utilization of the backbone layer and self-dilation processing module, respectively.

   - **Skip Connection Contribution:** An average skip contribution $\tilde{C}_i^s$ is computed by considering influences from $k$ previous layers (where $k = \min(i, N)$) via shortcut connections:

     $$\tilde{C}_i^s = \sum_{j=1}^{k} \left( \mathbb{E}[\mathbf{H}_i^r] \cdot D_{i-j,j}^s + C_{i-j} \right) \tag{17}$$

     where $\mathbb{E}[\mathbf{H}_i^r]$ is the mean output of the gating module governing shortcut auxiliary branch to layer $i$, and $D_{i-j,j}^s$ is the depth of the shortcut auxiliary branch processing module from layer $i - j$ to layer $i$.

   - **Final Effective Decision Depth:** The effective decision depth at the current backbone layer output is obtained by averaging the base depth and skip contributions:

     $$C_i = \frac{\tilde{C}_i^b + \tilde{C}_i^s}{k + 1} \tag{18}$$

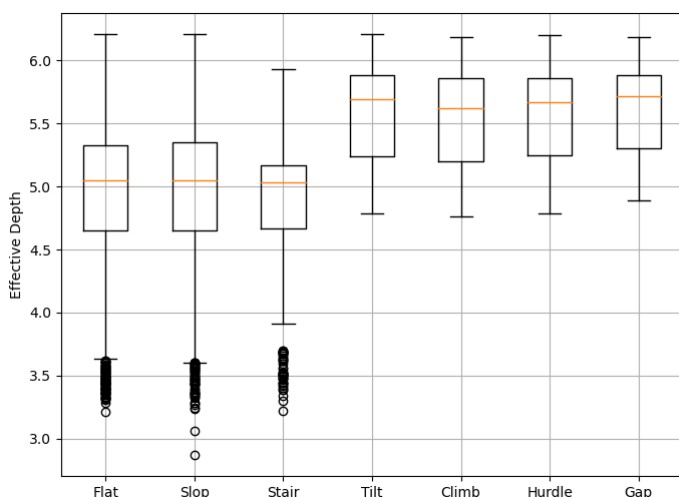

Figure S6: Boxplots for DPNet effective decision depth on the 7 quadrupedal locomotional tasks.

3. **Termination:** The process repeats iteratively until the effective decision depth $C_L$ for the final backbone feature $\mathbf{F}_L$ is computed.

The final value $C_L$ represents the overall processing complexity for a given input and is reported as the effective decision depth of the forward pass.

## G    EFFECTIVE DECISION DEPTH RESULTS ON QUADRUPEDAL LOCOMOTION TASKS

This section compares the effective decision depth of DPNet across different task types, examining the three walking tasks (Flat, Slope, Stair) versus the four parkour tasks (Tilt, Climb, Hurdle, Gap). The analysis reveals distinct depth utilization patterns between the two task categories.

For walking tasks, DPNet consistently employed shallower computational pathways, with median effective depths clustering around 5.0. The lower distribution boundaries were substantially lower, with first quartile values near 4.7 and lower whiskers extending to approximately 3.7. Additionally, all three walking tasks exhibited outliers with effective depths as low as 3.2, indicating instances where minimal computational depth sufficed.

In contrast, parkour tasks demonstrated systematically deeper pathway utilization. Median effective depths were consistently higher (around 5.7), while the first quartile (approximately 5.3) and lower whisker (around 4.8) values were significantly higher compared to walking tasks. This systematic upward shift across all distribution metrics confirms that DPNet adaptively selects deeper computational pathways for compositional parkour tasks while employing shallower pathways for atomic walking tasks.

## H    THE FINAL REWARDS OF DIFFERENT MINIHACK TASKS

The final individual MiniHack task rewards for different continual learning baseline algorithms (analyzed in Section 4.4 of the main paper) are presented in Table S3. Our proposed method consistently outperformed the continual learning baseline across most tasks, where the highest rewards reached 1.000. However, on Corridor-R2-V0, CorridorBattle-v0, and HideNSeek-v0, the rewards achieved were slightly lower than those of CLEAR. We attribute this to CLEAR's use of data replay to maintain individual task performance, an effective but hardware-intensive approach. Overall, our proposed DPNet design and progressive training technique effectively mitigated conflicts, thereby enhanced multi-task reinforcement learning performance on MiniHack.

Table S2: Direct training rewards of different MiniHack tasks.

| Task ID | Task name | Direct training rewards |
|---------|-----------|-------------------------|
| 1 | Room-Random-5x5-v0 | 0.845±0.003 |
| 2 | Corridor-R2-v0 | -0.862±0.035 |
| 3 | Room-Dark-5x5-v0 | 0.781±0.007 |
| 4 | Corridor-R3-v0 | -0.748±0.126 |
| 5 | Room-Monster-5x5-v0 | 0.677±0.021 |
| 6 | CorridorBattle-v0 | 0.018±0.002 |
| 7 | Room-Trap-5x5-v0 | 0.816±0.003 |
| 8 | HideNSeek-v0 | 0.019±0.003 |
| 9 | Room-Ultimate-5x5-v0 | 0.565±0.023 |
| 10 | HideNSeek-Lava-v0 | 0.026±0.002 |

Table S3: The final rewards of different individual tasks in MiniHack.

| Tasks | EWC (J. et al., 2017) | P&C (S. et al., 2018) | CLEAR (R. et al., 2019) | SANE (S. et al., 2022) | Ours |
|-------|-----------------------|-----------------------|-------------------------|------------------------|------|
| Room-Random-5x5-v0 | 0.897±0.000 | 0.884±0.000 | 0.816±0.003 | 0.819±0.034 | 1.000±0.000 |
| Corridor-R2-v0 | 0.049±0.062 | 0.224±0.018 | 0.505±0.049 | -0.070±0.140 | 0.480±0.014 |
| Room-Dark-5x5-v0 | 0.588±0.000 | 0.850±0.016 | 0.633±0.019 | 0.892±0.023 | 0.744±0.003 |
| Corridor-R3-v0 | -0.468±0.001 | -0.289±0.001 | -0.596±0.003 | -0.910±0.004 | -0.347±0.001 |
| Room-Monster-5x5-v0 | 0.931±0.002 | 0.812±0.009 | 0.892±0.008 | 0.963±0.003 | 1.000±0.000 |
| CorridorBattle-v0 | -0.320±0.001 | -0.039±0.000 | 0.357±0.049 | 0.008±0.161 | -0.141±0.006 |
| Room-Trap-5x5-v0 | 0.863±0.002 | 0.781±0.000 | 0.780±0.032 | 0.999±0.000 | 1.000±0.000 |
| HideNSeek-v0 | 0.624±0.003 | -0.014±0.000 | 0.698±0.027 | 0.731±0.002 | 0.298±0.010 |
| Room-Ultimate-5x5-v0 | 0.694±0.007 | 0.809±0.002 | 0.636±0.003 | 0.853±0.001 | 0.897±0.010 |
| HideNSeek-Lava-v0 | 0.597±0.007 | 0.026±0.002 | 0.664±0.015 | 0.731±0.009 | 0.847±0.002 |
| Final Average Reward | 0.445±0.001 | 0.4045±0.003 | 0.538±0.005 | 0.502±0.003 | **0.578**±0.001 |