# OpenReview forum: "Mitigating Conflicts in Multi-Task Reinforcement Learning via Progressively-Trained Dynamic Policy Network"
_ICLR.cc/2026/Conference — ICLR 2026 Conference Withdrawn Submission_

### Official Review · Reviewer_odXW · 2025-10-31

**Soundness:** 1
**Presentation:** 1
**Contribution:** 1
**Rating:** 0
**Confidence:** 4

**Summary:**

The paper deals with the problem of multi-task RL, where there exists inter-task conflicts. To mitigate
this, the paper introduces a dynamic policy network that incorporates diverse computational pathways of varying depths, along with gating modules. Moreover, a progressive training technique is developed to further mitigate the conflicts among
tasks. The proposed method is demonstrated on quadrupedal locomotion tasks against some baseline methods.

**Strengths:**

- the investigated problem, i.e., multitask RL, is important.

**Weaknesses:**

- The paper is poorly written and poorly structured. For instance, experimental results are presented in the introduction section. A lot of concepts are introduced without clear definition or description, e.g., backbone feature, action frequency, etc.

- The proposed method is not well-motivated. For instance, section 3.3 progressive training is motivated using Figure 3, which lacks clear information or evidence.

- The experimental results are weak, without clear evidence that the proposed method outperforms existing multitask RL methods.

**Questions:**

- Is the dynamic policy network only applicable to multilayer perceptron architecture? what if the backbone policy network is transformer-based?

---

### Official Review · Reviewer_QAfV · 2025-11-01

**Soundness:** 2
**Presentation:** 2
**Contribution:** 2
**Rating:** 4
**Confidence:** 3

**Summary:**

This paper tackles the problem of negative transfer, or inter-task conflict, in multi-task reinforcement learning (RL). The authors propose two main contributions:

1. A novel network architecture that introduces "auxiliary branches" (shortcuts) and "self-dilation" pathways to the network's backbone. These branches create diverse computational pathways of varying depths. Gating modules are used to selectively activate these pathways, allowing the network to dynamically choose shallower paths for simpler tasks and deeper paths for more complex ones.
2. A novel training technique designed to mitigate a specific hypothesized source of conflict: competition over low-frequency actions between simple "atomic" tasks and complex "compositional" tasks. The technique first trains on atomic tasks with astrong smoothness rewardto learn stable, low-frequency actions. It then trains on compositional tasks, reducing the smoothness reward and using a continual learning method (like EWC) to preserve the previously learned low-frequency foundations.

The authors evaluate their combined approach on a 7-task quadrupedal locomotion suite and 10 MiniHack navigation tasks. In both benchmarks, DPNet outperforms baseline approaches.

**Strengths:**

- This work is highly significant for multi-task and continual RL. The "action frequency conflict" hypothesis provides a new, concrete way to analyze and address negative transfer.
- The empirical evidence demonstrates significant performance improvements over all baselines in multiple environments.
- The paper is well-structured.

**Weaknesses:**

- The motivation for the specific design of the processing module ($S$) and gating module ($G$) is unclear. The diagram (Figure 2) is highly reminiscent of a MoE layer, yet the paper uses an element-wise product rather than the standard gated summation. The paper would be stronger if it justified this specific design choice, perhaps with an ablation study comparing the performance of this module to a standard MoE layer.
- The distinction between the "Backbone Module" ($B_i$) and the "Processing Module" ($S_{i,m}$) is poorly defined. From Figure 2, their primary difference appears to be that Processing Module has a shortcut path. It is unclear if they differ in internal architecture or parameter count. The paper should clarify the design rationale: Why is a separate Backbone Module necessary? Why do only Processing Module utilizes shortcut connection?
- The comparison against multi-task RL baselines in Table 3 needs more implementation details for reproducibility and fairness. Furthermore, several baselines were adapted from different enviroments, and the paper should detail these adaptations. For example, how were the "context-based representations" from the original CARE paper implemented for the locomotion environment?
- The most substantial performance gains appear to stem from the progressive training technique, not the DPNet architecture alone. As shown in Table 2, the full DPNet architecture (under the "Pretrain" setting) provides only a modest gain (8.0 Avg.TLR) over the MLP baseline (7.6 Avg.TLR). The significant jump to 9.2 Avg.TLR only occurs when progressive training is applied. This confounds the main results in Table 3, as all baseline architectures were also evaluated using this same progressive training technique. It is unclear if DPNet is a superior architecture or if progressive training is just a powerful technique that universally boosts all models. To properly isolate the architectural contribution, a new ablation study is needed comparing all architectures (DPNet, MoE, CARE, etc.) under the same baseline "Pretrain" setting.

**Questions:**

See Weaknesses.

---

### Official Review · Reviewer_MAwb · 2025-11-01

**Soundness:** 2
**Presentation:** 1
**Contribution:** 2
**Rating:** 2
**Confidence:** 4

**Summary:**

This paper introduces an approach to mitigating conflicts in multi-task reinforcement learning (RL) by combining a dynamic policy network (DPNet) design with a progressive training technique. The idea of the method is to incorporate multiple computational pathways by varying the depth + gating modules to allow the selection of suitable pathways for different tasks (of different complexity). Progressive training is a curriculum training that uses a specific training order + continual learning to train policies on harder tasks.

**Strengths:**

The method shows significant improvement on the continual learning benchmark - MiniHack.

**Weaknesses:**

1. [Writing] The writing quality is not up to the mark for a publication in a top ML conference. The flow of the paper is not sequential, and as a reader, I had to go back and forth between different sections of the paper to understand it. I believe that this paper needs a rewrite. Some of the issues that I've come across are mentioned below:

	- The "Action Frequency" section in the preliminary is very abrupt.
	- The paper lacks a preliminary section on continual learning, which I believe would be helpful for the reader as a context.
	- [Minor comment] Citations are formatted in a weird manner where it shows {last initial., et al} instead of {last name., et al}. This is a minor comment.
	- Some related works sections, like "Compositional Networks for Multi-Task Learning" and "Continual Reinforcement Learning," lack contrast of DPNet with other existing works. The main job of the related works section is to *compare* and *contrast* the current work with prior works and contextualize it.

2. [Claims of progressive training]: The progressive training that is claimed to be proposed in this work is not a new strategy. In fact, it has been addressed for quadruped locomotion in Eurekaverse [1], where an environment curriculum was adopted during policy training.

3. [Experiments: Environment] Metaworld suite of tasks is a common testbed for multi-task RL, and comparison on them should be provided in the paper.

4. [Experiments: Baselines] Specifically for the Quadruped tasks - it is unclear to me how much of the contribution is coming from DPNet and how much is from the curriculum training? I would like to see an experiment without DPNet and only the curriculum.

5. The validity of the Quadruped experiments for Sim2Real transfer is unclear and hasn't been shown.

----
**References:**

 [1]  Eurekaverse: Environment Curriculum Generation via Large Language Models, William Liang et al., CoRL 2024

**Questions:**

6. How much of a reduction in training/inference speed is observed for policies trained with DPNet?

7. [No experiment needed] How do DPNet class of models differ from gradient surgery type of methods like PCGrad [2]?

----
**References**

[2] Gradient surgery for multi-task learning, Tianhe Yu et al., NeurIPS (2020).

---

### Note · Authors · 2026-01-13

I have read and agree with the venue's withdrawal policy on behalf of myself and my co-authors.